# Applying Physiologically Based Pharmacokinetic Modeling to Interpret Carbamazepine’s Nonlinear Pharmacokinetics and Its Induction Potential on Cytochrome P450 3A4 and Cytochrome P450 2C9 Enzymes

**DOI:** 10.3390/pharmaceutics16060737

**Published:** 2024-05-30

**Authors:** Xuefen Yin, Brian Cicali, Leyanis Rodriguez-Vera, Viera Lukacova, Rodrigo Cristofoletti, Stephan Schmidt

**Affiliations:** 1Center for Pharmacometrics and Systems Pharmacology, Department of Pharmaceutics, College of Pharmacy, University of Florida, Orlando, FL 32827, USA; xuefenyin@ufl.edu (X.Y.); bcicali@ufl.edu (B.C.); leyanisrv80@gmail.com (L.R.-V.); 2Simulations Plus, Lancaster, CA 93534, USA; viera.lukacova@simulations-plus.com

**Keywords:** physiologically based pharmacokinetic (PBPK) modeling, Drug–Drug Interaction (DDI), carbamazepine, Carbamazepine-10,11-epoxide, induction, cytochrome P450 3A4 (CYP3A4), cytochrome P450 2C9 (CYP2C9)

## Abstract

Carbamazepine (CBZ) is commonly prescribed for epilepsy and frequently used in polypharmacy. However, concerns arise regarding its ability to induce the metabolism of other drugs, including itself, potentially leading to the undertreatment of co-administered drugs. Additionally, CBZ exhibits nonlinear pharmacokinetics (PK), but the root causes have not been fully studied. This study aims to investigate the mechanisms behind CBZ’s nonlinear PK and its induction potential on CYP3A4 and CYP2C9 enzymes. To achieve this, we developed and validated a physiologically based pharmacokinetic (PBPK) parent–metabolite model of CBZ and its active metabolite Carbamazepine-10,11-epoxide in GastroPlus^®^. The model was utilized for Drug–Drug Interaction (DDI) prediction with CYP3A4 and CYP2C9 victim drugs and to further explore the underlying mechanisms behind CBZ’s nonlinear PK. The model accurately recapitulated CBZ plasma PK. Good DDI performance was demonstrated by the prediction of CBZ DDIs with quinidine, dolutegravir, phenytoin, and tolbutamide; however, with midazolam, the predicted/observed DDI AUC_last_ ratio was 0.49 (slightly outside of the two-fold range). CBZ’s nonlinear PK can be attributed to its nonlinear metabolism caused by autoinduction, as well as nonlinear absorption due to poor solubility. In further applications, the model can help understand DDI potential when CBZ serves as a CYP3A4 and CYP2C9 inducer.

## 1. Introduction

Carbamazepine (CBZ), a tricyclic anticonvulsant, is a widely used medication for treating various types of epilepsy, including partial seizures, generalized tonic–clonic seizures, and mixed seizures [1]. CBZ is also effective in managing trigeminal neuralgia, a type of facial pain, and it is increasingly being used in the treatment of various psychiatric disorders, such as bipolar disorder [2]. It is a highly lipophilic and low-solubility drug, classified under the Biopharmaceutics Classification System (BCS) class II [3]. CBZ was suggested to be taken with meals despite its only mild food effect [4,5,6]. Upon entering systemic circulation, 70–80% of CBZ binds to plasma protein [7,8,9]. CBZ undergoes almost complete metabolism through several pathways, with the CBZ epoxide–diol pathway being the major one. This pathway involves the action of enzymes including cytochrome P450(CYP)3A4, CYP3A5, and CYP2C8, which biotransform CBZ into the pharmacologically active metabolite Carbamazepine-10,11-epoxide (CBZ-E) [10,11,12,13]. Epoxide Hydrolase 1(EPHX1) then further metabolizes CBZ-E into the inactive carbamazepine-10,11-transdihydrodiol (CBZ-diol) [14]. Other pathways include CBZ hydroxylation, catalyzed by CYP3A4 and CYP2B6 as major responsible enzymes [11], and the formation of CBZ-N-glucuronide via UGT2B7 [15]. Less than 2% of an oral dose is excreted as unchanged CBZ in urine [16,17].

CBZ displays nonlinear pharmacokinetics (PK) in healthy subjects following a single oral dose, but the root cause(s) is/are not yet fully understood. Several studies [6,18,19] have attempted to identify the underlying mechanism by analyzing the relationship between the dose and the product of the elimination rate constant (Ke) and the area under the plasma concentration curve extrapolated to time infinity (AUC_0-inf_). Ke × AUC_0-inf_ represents bioavailability(F) × dose, assuming a constant apparent volume of distribution. Cotter et al. [18] reported that Ke increases with increasing CBZ dosage (200 to 900 mg, tablet) and that Ke × AUC_0-inf_ has a linear relationship with the dose, leading them to conclude F was independent of the dose. This is line with the findings of Gerardin et al. [19] for 100 to 600 mg oral tablets but inconsistent with the results of Levy et al. [6] for 3 to 9 mg/kg doses administered as propylene glycol solution. Levy et al. observed no significant difference in half-life and noticed a parallel elimination rate as well as a linear relationship between Ke × AUC_0-inf_ and dose in four of six subjects, though two of six subjects showed a nonlinear PK, which may be due to CBZ precipitation at a higher dose. The relative F of the tablet compared to the solution was 79%, with a range of 56–109% (6 mg/kg), and it increased after food intake, although the difference was not significant. Based on these observations, in conjunction with the low water solubility of CBZ and the rapid disintegration of commercial tablets (less than 30 s with no agitation), Levy et al. proposed that CBZ has a dissolution-rate-limited absorption. Due to these contradicting conclusions, the underlying mechanism(s) of CBZ’s nonlinear PK remain(s) unclear, and further studies are needed to come up with an unambiguous explanation.

CBZ is a well-known inducer of various drug-metabolizing enzymes. The U.S. Food and Drug Administration (FDA) classifies CBZ as a strong inducer of CYP3A4 and CYP2B6 enzymes, as well as a weak inducer of CYP2C9 [20]. Repeated administration of CBZ can upregulate these enzymes, resulting in an increase in CBZ’s metabolism as it undergoes metabolism by CYP3A and CYP2B6. CBZ also induces many other drug enzymes, including, but not limited to, CYP3A5 and UGT2B7, making it prone to Drug-Drug Interactions (DDIs). For instance, significant DDIs may occur when CBZ is taken together with phenytoin (PHT), another commonly used antiepileptic drug primarily metabolized by CYP2C9, particularly in patients whose seizures cannot be controlled with monotherapy. CBZ induces CYP2C9, increasing the elimination rate of PHT, while PHT is a CYP3A4 inducer, which accelerates the biotransformation rate of CBZ. Given the narrow therapeutic range of both drugs [21,22], such interactions may have significant clinical consequences and result in undertreatment. In addition, co-administration of CBZ with levonorgestrel, a commonly used active moiety in many hormonal contraceptive formulations, which is metabolized by CYP3A4, can reduce levonorgestrel concentrations, potentially leading to contraceptive failure due to inadequate exposure needed for desired efficacy [23]. Therefore, it is essential to understand any potential drug interactions before prescribing CBZ and to consider alternative medications to prevent adverse drug interactions.

CBZ is expected to play an increasingly important role in CYP3A clinical DDI studies. CYP3A is one of the most important drug-metabolizing enzymes in the body, and it is involved in the metabolism of many drugs [24]. Although both CBZ and Rifampin are well-recognized strong CYP3A clinical index inducers, Rifampin’s use in DDI studies has faced concerns, particularly in healthy volunteer populations, due to the presence of N-Nitrosamine impurities that exceed the acceptable limits [25]. Therefore, researchers are seeking alternatives to Rifampin, and, after thorough comparison, CBZ has been identified as one of the three most promising alternatives, alongside phenytoin and lumacaftor. CBZ’s extensive clinical DDI experience, the availability of physiologically based pharmacokinetic (PBPK) models, and the established correlation between CBZ-induced CYP3A activity and the levels of the endogenous biomarker 4β-hydroxycholesterol (4βOHC), make it an attractive option for CYP3A-related DDI studies. By studying CBZ’s effects on the PK of new drugs, researchers can enhance their understanding of the CYP3A-related DDIs of these new drugs and more accurately anticipate potential interactions with other CYP3A inducers.

A comprehensive carbamazepine PBPK model can be utilized to improve our understanding of CBZ’s PK behavior, improve clinical trial designs, and secure regulatory approval. Regulatory agencies, such as the FDA and the European Medicines Agency (EMA), recognize the importance of PBPK modeling and recommend its use in studying drugs’ PK within the whole body and their potential DDIs [26,27]. By estimating the impact of DDIs on PK in plasma and other tissues, PBPK modeling enables the evaluation of alternative dosing strategies to minimize the side effects of interactions. Moreover, PBPK simulations have the potential to waive the need for additional clinical studies, saving valuable time and resources as well as reducing unnecessary burden on the patients, while still providing essential information about the safety and efficacy of new drugs when combined with drugs like CBZ [28]. Thus, the goal of this study is to apply the PBPK modeling approach to explore the mechanism behind CBZ’s nonlinear PK and its induction potential on other drugs metabolized by CYP3A4 and CYP2C9 enzymes. The study was conducted in four steps: (1) developing a CBZ parent–metabolite (P-M) PBPK model for single-dose CBZ; (2) describing the autoinduction effect of CBZ; (3) assessing CBZ DDI potentials on CYP3A4 and CYP2C9 pathways using CYP3A4 and CYP2C9 prototypical victim drugs; and (4) investigating the mechanisms of CBZ’s nonlinear PK.

## 2. Materials and Methods

### 2.1. Software

The CBZ P-M PBPK model was developed using GastroPlus^®^ version 9.8.2. The DDI Module in GastroPlus^®^ was used to capture the autoinduction effect of CBZ after multiple-dose administration and to predict DDIs through dynamic simulation. Parameter optimization and sensitivity analysis (PSA) were carried out in GastroPlus^®^. Clinical study data were digitized from the scientific literature using WebPlotDigitizer-4.5. The PK parameters for single-dose regimens were calculated using the PKPlus ™ Module in GastroPlus^®^, while PK parameters for multiple-dose regimens were calculated using Phoenix^®^ WinNonlin^®^ version 8.4 (Certara USA, Inc., Princeton, NJ, USA), and the goodness-of-fit plots were created with R 4.1.3 (the R foundation for Statistical Computing, Vienna, Austria 2021).

### 2.2. Clinical Data

Plasma or serum concentration–time profiles of CBZ and CBZ-E were obtained from published clinical studies. The inclusion criteria for the studies were the following: (1) clearly defined administration route and provided dose; (2) contained CBZ and/or CBZ-E’s observed plasma or serum concentration–time curve plots and/or data; and (3) conducted in healthy volunteers. The collected clinical studies were further divided into two sets: a training dataset for model development and a test dataset for model verification. The training dataset included studies that covered a wide range of doses, various formulations of CBZ, single- and multiple-dose regimens, and plasma or serum concentrations of CBZ-E in order to provide insight into the PK processes that were implemented. Details of the clinical trials used can be found in Appendix A, indicating whether they were assigned to the training or test dataset.

### 2.3. Model Development and Verification

The workflow of the CBZ P-M PBPK model’s development and verification is presented in Figure 1.

CBZ drug-specific parameters on absorption, distribution, metabolism, and excretion (ADME) were determined through a three-pronged approach: (1) reported values that best described the observed drug PK profiles, tested via a PSA-guided approach; (2) optimized values for the parameters sensitive to model predictions if they were not available or if reported values failed to accurately capture the drug’s PK characteristics; and (3) default values predicted using ADMET Predictor^®^ v10.3.0.0 if parameters were neither sensitive nor available.

Virtual individuals were generated using the Population Estimates for Age-Related (PEAR) Physiology™ population analysis module in GastroPlus^®^. The following demographic factors were taken into account: (1) the mean age of the study subjects, (2) the mean body weight or BMI of the subjects, (3) the sex of the subjects (a male PBPK model was utilized if the number of males equaled or exceeded the number of females, or if sex was not mentioned), (4) the fasting or fed state of the subjects, and (5) the specified water consumption in the clinical study protocols. In instances where the source publication did not provide any participant information, a standard individual was presumed to be a 30-year-old, 75 kg European male who consumed 250 mL of water.

In the Advanced Compartmental Absorption and Transit (ACAT^TM^) model, the absorption scaling factors (ASFs) were calculated using the default Opt logD Model SA/V 6.1. Paracellular permeability was included in the permeability model and calculated using the default Zhimin method due to the low molecular weight of CBZ and CBZ-E. The systemic distribution of CBZ and CBZ-E was described with a perfusion-limited tissue model for all tissues, and the tissue-to-plasma partition coefficients (Kps) were calculated using the default Lukacova (Rodgers-single) algorithm [29].

The CBZ P-M PBPK model for single-dose regimens was developed through a step-by-step process. The CBZ-E PBPK model was created first using CBZ-E PK data from two different clinical studies involving the administration of CBZ-E to healthy subjects, with a dosage of 100 mg in suspension and an enteric-coated tablet. The elimination pathways for the CBZ-E model included EPHX1 hepatic clearance (CL_hepatic_) described by linear intrinsic clearance and renal elimination of the unchanged drug (CL_renal,filt_). To determine the CL_hepatic_ of CBZ-E, a back-calculation approach was used based on the clinical study by Tomson et al. [30]. Non-compartment analysis (NCA) was performed in the PKPlus^TM^ module to obtain individual CL_sys_/F (where CL_sys_ represents total systemic clearance of CBZ-E), and the mean value of 5.28 L/h was calculated. CL_hepatic_ was then calculated using the equation CL_hepatic_ = CL_sys_/F × F_pre_ − CL_renal,filt_ = 4.86 L/h. Here, CL_renal,filt_ was used as previously reported [16]. F_pre_ = (1 − CL_hepatic_/(Q_h_ × R_bp_)) × Fa, where Q_h_ represents the hepatic blood flow rate and Fa represents the fraction of the drug absorbed with a simulated value of 0.9995. The formulation parameters for CBZ-E in solution and suspension used the default values, while the ones for enteric-coated tablets were optimized using the Weibull function. The model did not include the CBZ-E degradation mechanism in gastric fluid, despite evidence of its instability in such an environment [31,32]. This was primarily because (1) aside from one CBZ-E study examining degradation [31], the remaining studies included co-administered antacids or an enteric-coated formulation, and (2) the focus of this model was on CBZ administration, where CBZ-E degradation in gastric fluid is not relevant. To account for CBZ-E degradation in gastric fluid in the one study [31] not including antacid co-administration, the dose was adjusted to represent only the amount of CBZ-E not degraded in the stomach before emptying into the small intestine. The CBZ-E PBPK model was validated based on four clinical studies of an oral single-dose administration of CBZ-E in a fasted condition, covering a 50 to 200 mg dosing range, with all available drug formulations (suspension, solution, and enteric-coated tablet). Please refer to Appendix A for more information.

The CBZ P-M PBPK model for CBZ single dose was developed by integrating the previously verified CBZ-E PBPK model with the CBZ parent PBPK model. Seven datasets from five clinical studies were used to train the CBZ parent PBPK model, involving healthy individuals receiving single doses of CBZ at doses ranging from 10 to 400 mg and in various formulations (please refer to Appendix A). The study with CBZ P-M iv administration was used to parameterize CBZ systemic distribution and elimination. The elimination pathways for the CBZ parent model include (1) biotransformation to CBZ-E by CYP2C8, CYP3A4, and CYP3A5, (2) formation of hydroxylated CBZ metabolites by CYP2B6, CYP3A4, and of CBZ -N-glucuronide via UGT2B7, (3) unspecified metabolic processes through liver clearance (CL_int,liver,unbound_, contributing 56% in total) encompassing both the studied enzymes with minimal impact on CBZ elimination and all unstudied metabolic pathways, and (4) unchanged drug excretion into urine (CL_renal,filt_). Only enzymes CYP3A4 and UGT2B7 were incorporated in the gut due to the low intestinal expression of the other enzymes [33,34]. The metabolism pathways were implemented using Michaelis–Menten Kinetics, apart from CL_int,liver,unbound_, which was described as linear intrinsic clearance. Figure 2B provides an overview of the CBZ metabolism process. The systemic disposition of CBZ was initially calibrated using CBZ exposure data from the Gerardin 1990 study [35] (subject 1 received 10 mg of CBZ through iv infusion), and the CL_int,liver,unbound_ was further refined using data from the Meyer 1992 study [36]. The formulation parameters were incorporated to simulate CBZ oral administration. The CBZ oral drug formulations included solution, suspension, immediate-release (IR) tablets, control-release (CR) tablets, and CR capsules. The formulation parameters for each form were introduced separately into the CBZ parent model, such that (1) the GastroPlus^®^ default values were used for solution and suspension, (2) the fixed particle size distribution information based on the literature [37,38,39,40] was used for IR tablets, and (3) an optimized Weibull function was used to describe CBZ dissolution from CR tablets and CR capsules in the gastrointestinal tract. The CBZ P-M PBPK model for a single-dose regimen was validated using 27 clinical studies covering a wide range of dosages (10 to 800 mg) and different drug formulations (iv infusion, suspension, solution, IR tablets, CR tablets, and CR capsule). Please refer to Table 1 and Appendix A for more information.

The CBZ P-M PBPK model, initially developed for a single-dose regimen, was expanded to account for the impact of CBZ autoinduction on CBZ plasma concentration after multiple doses. The maximum effect model was employed for CYP3A4, CYP3A5, CYP2C8, and CYP2B6 enzymes, which are represented by green arrows in Figure 2C. The model was parameterized using a clinical study by Ji et al. [41]. The simulations were completed in the DDI module in GastroPlus^®^. The half-maximal effective concentration (EC_50_) and the maximal effective values (E_max_) for all enzymes, except for CYP3A4, were obtained from in vitro experiments based on mRNA analysis (Table 2). For CYP3A4 induction, the EC_50_ value of 22 µM, also a reported CYP2C8 value from mRNA analysis [42], was used. This aligns well with an average value of 22.13 µM obtained from CYP3A4 enzyme activity measurements (Appendix A) and 20 µM utilized in a previously published CBZ PBPK model [43]. The E_max_ value for CYP3A4 was fitted against clinical data reported by Ji et al. [41]. Five additional multiple-dose clinical studies were used to validate the final model. Please refer to Appendix A for more information.

### 2.4. Model Evaluation

Model performance was evaluated as follows.

(1) By visually comparing the predicted plasma concentration–time profiles to the observed ones.

(2) By comparing the predicted with the observed areas under the plasma concentration–time curve (AUC) and maximum plasma concentration (C_max_) values. AUC values were calculated from the time of drug administration to the time of the last concentration measurement (AUC_last_) and to the time of infinity (AUC_0-inf_). Predictions within 80–125% of the observed values were considered accurate, and predictions within 2-fold of the observed values were considered adequate.

(3) By comparing predicted plasma concentration values to the corresponding observed values. For a quantitative description of the model’s performance, the absolute average-fold error (AAFE) of predicted plasma concentrations was calculated according to Equation (1). AAFE values < 2 were considered to be adequate model performance metrics.
(1)$$AAFE=10^{\frac{\sum_{i=1}^{n}\left|log\frac{Predicted}{Observed}\right|}{n}}$$

### 2.5. Parameter Local Sensitivity Analysis

The sensitivity of CBZ exposure to PBPK model input parameters was evaluated for parameters that (1) were optimized, (2) showed high impact on model prediction, and 3) were related to optimized parameters. A local sensitivity analysis was conducted by varying each parameter of interest by +/− 20% (within a 0.8- to 1.2-fold range) in 9 logarithmic steps, while keeping all other parameters constant. The sensitivity of a parameter was calculated based on Equation (2), reported by Hanke et al. [44]. In the equation, p represents the original model parameter value, ∆p shows the relative change in parameter value, AUC_0−inf_ indicates the simulated AUC_0−inf_ value with the original parameter value, and ∆ AUC_0−inf_ denotes the change of AUC_0−inf_ based on the relative variation of one parameter. A sensitivity of +1.0 means that a 10% increase of the examined parameter results in a 10% rise of the simulated AUC_0−inf_. Results of the sensitivity analysis can be found in the Appendix A.
(2)$$Sensitivity=\frac{∆{AUC}_{0-inf}}{{AUC}_{0-inf}}\times \frac{p}{∆p}$$

### 2.6. DDI Application and Evaluation

The CBZ P-M PBPK model was coupled with the built-in PBPK models of victim drugs in GastroPlus^®^ to assess CBZ DDI potentials on enzyme CYP3A4 and CYP2C9. Specifically, dolutegravir and quinidine model versions 1 and midazolam model version 2 were used as CYP3A4 victim drugs, while tolbutamide model version 1 and phenytoin (in-house model [45]) were used as CYP2C9 victim drugs (illustrated in Figure 2, part C). In these clinical studies, CBZ was first titrated to a steady state, resulting in full induction of CYP3A4 and CYP2C9 before the co-administration of a single dose of victim drugs. The CBZ DDI studies with midazolam and tolbutamide have no observed plasma concentration vs. time profiles in the literature, and the quality of prediction was evaluated only through comparison with the reported observed AUC_last_ and C_max_ ratios. Information on all utilized DDI studies, including study protocols, demographics, and references, can be found in Appendix A. All enzyme induction parameters were used as validated against CBZ multi-dose studies, as described in Section 2.3 (model development and verification). In addition, the E_max,CYP2C9_ value is derived from the mRNA increase [42], and the value of EC_50,CYP2C9_ is set to 22 µM, which is equal to that of EC_50,CYP2C8_ and EC_50,CYP3A4_.

The DDI predictive performance of the model was evaluated through comparison of the predicted versus observed victim drug plasma concentration–time profiles during co-administration. In addition, the predicted DDI C_max_ ratios (Equation (3)) and DDI AUC_last_ ratios (Equation (4)) were compared to the respective observed ratios. As a quantitative measure of the prediction accuracy of each DDI interaction, the success criteria for the DDI C_max_ ratio and DDI AUC ratio were calculated based on Guest criteria [46]. The upper limit and the lower limit were calculated based on Equations (5)–(7), accounting for 20% of intraindividual variability, whereas R_obs_ represents the observed DDI C_max_ ratio or DDI AUC_last_ ratio.
(3)$$DDI\;C_{max}\;ratio=\frac{C_{max,coadministration}}{C_{max,alone}}$$
(4)$$DDI\;{AUC}_{last}\;ratio=\frac{{AUC}_{last,coadministration}}{{AUC}_{last,alone}}$$
(5)$$Upper\;limit=R_{obs}\times Limit$$
(6)$$Lower\;limit=R_{obs}/Limit$$
(7)$$Limit=\frac{1.25+2\left(R_{obs}-1\right)}{R_{obs}}$$

### 2.7. CBZ Nonlinear PK Exploration

The CBZ P-M PBPK model was utilized to investigate mechanisms responsible for the nonlinear PK through simulations of single and multiple doses ranging from 50 mg to 800 mg, based on the Gerardin 1976 clinical study [19]. As illustrated in Figure 2A, CBZ needs to undergo release (or disintegration), dissolution, and absorption and then escape first-pass metabolism in the gut and liver before becoming available in systemic circulation. Nonlinearity can occur at several of these stages. To explore the underlining mechanisms, several relationships were analyzed, including AUC/dose versus dose, total systemic clearance (CL) versus dose, dissolved percentage (Dis%) versus dose, absorbed percentage (Fa%) versus dose, portal vein entry percentage (FDp%) versus dose, and F versus dose. In addition, the CBZ concentrations in the enterocytes and the liver were compared with Km values for CYP3A4 and CYP3A5, which are two major enzymes in CBZ metabolism.

## 3. Results

### 3.1. CBZ-E PBPK Model Building and Performance

The CBZ-E drug-dependent parameters and formulation parameters are listed in Table 1. The local sensitivity analysis for the CBZ-E PBPK model indicated that the plasma unbound fraction (fu_p_) and liver clearance (CL_liver_) are the key model parameters, with sensitivity values of −0.83 and −0.81, respectively, shown in Appendix A.

In the simulation of Sumi’s 1987 clinical study [31] (Appendix A), the given dose was corrected from 150 mg to 121.5 mg to account for CBZ-E degradation in gastric fluid, as this study did not include antacid co-administration, while other studies used for model establishment and verification did. To be consistent with other clinical studies, Equation (8) is used to correct the given dose. In the equation, 0.81 is the bioavailability parameter reported in the literature [31] obtained by comparing the AUC parameters following oral administration of CBZ-E 150 mg solution with or without 30 mL of an antacid.
(8)$${Dose}_{corrected}=Dose\times 0.81$$

The model accurately predicted the CBZ-E plasma concentration–time profiles observed in healthy subjects (Appendix A), with 70%, 80%, and 60% of simulated AUC_0-inf_, AUC_0-t_, and C_max_ values within 80–125% of the observed values, respectively, and all predicted PK parameters within two-fold of the observed values (Appendix A). In total, 10/10 of the AAFE values are within two-fold range (Appendix A).

**Table 1 pharmaceutics-16-00737-t001:** CBZ and CBZ-E PBPK model inputs.

Parameter		Model Value	Literature Value
**Carbamazepine drug-dependent parameters**			
MW (g/mol)		236.27	-
logP		2	1.51 [47], 2.45 [48], 2.19 [49], 2.29 [50], 2.93 [51]
fu_p_ (%)		22.5	22.5–25.8 [52], 26–29.8 [8]
Blood: plasma concentration ratio (R_bp_)		0.9	1.06 ± 0.21 [53], 0.9 ± 0.11 [54], 1.36 ± 0.10 [54]
pKa		10.86	10.86 (ADMET Predictor v.10.3.0.0), 11.83 [55], 14 [47]
CYP2C8 → CBZ-E	K_m_ (µM)	757	757 [56]
	V_max_ (nmol/min/nmol CYP)	0.669	0.669 [56]
CYP3A4 → CBZ-E	K_m_ (µM)	248	119 [56], 248 [10], 442 [13], 630 [57]
	V_max_ (pmol/min/pmol CYP)	0.75 ^1^	1.17 [56], 4.87 [10], 1.37 [13], 5.3 [57]
CYP3A5 → CBZ-E	K_m_ (µM)	2300	2300 [10], 338 [57]
	V_max_ (pmol/min/pmol CYP)	10	10 [10], 5.98 [57]
CYP2B6 → OH-CBZ	K_m_ (µM)	420	420 [11]
	V_max_ (pmol/min/pmol CYP)	0.429	0.429 [11]
CYP3A4 → OH-CBZ	K_m_ (µM)	282	282 [11]
	V_max_ (pmol/min/pmol CYP)	0.164	0.164 [11]
UGT2B7 → CBZ-glu	K_m_ (µM)	214	214 [15]
	V_max_ (pmol/min/mg)	0.79	0.79 [15]
CL_int,liver,unbound_ (L/h)		3.316 ^2^	-
CL_renal,filt_ (L/h)		0.0084	0.0084 [16]
Peff (cm/s × 10^−4^)		4.3	4.3 ± 2.7 [58,59]
Aqueous solubility (mg/mL, pH)		0.127 (6.5)	0.12(6.8) [37], 0.127(6.5) [60], 0.214(1) [61], 0.26(1) [39]
Solubility (mg/mL, SGF at pH = 1.2 at 0 mM)		0.236	0.236 [60]
Solubility (mg/mL, FaSSIF at pH = 6.8 at 3 mM)		0.283	0.132 [60], 0.234 [62], 0.24 [3], 0.27 [63], 0.283 [60], 0.31 [63]
Solubility (mg/mL, FeSSIF at pH = 6.8 at 15 mM)		0.52	0.343 [62], 0.47 [63], 0.52 [63]
Diffusion Coefficient (cm^2^/s × 10^5^)		0.86	ADMET Predictor v.10.3.0.0
**CBZ formulation parameters**			
**Solution and suspension**			
Particle density (g/mL)		1.2	GastroPlus^®^ default value
Mean Particle Radius (µm)		25	GastroPlus^®^ default value
Particle radius standard deviation		0	GastroPlus^®^ default value
Particle radius bin#		1	GastroPlus^®^ default value
**IR tablet**			
Particle density (g/mL)		1.5	1.5 [37]
Mean Particle Radius (µm)		60	75 [37], 1–40 [38], 62.5–100 [39], 7.5–168 [40]
Particle Radius standard deviation		20	20 [37]
Particle radius bin#		5	5 [37]
**CR tablet ^3^**			
T (time lag) (h)		0.5	Optimized value
Max (total release) (%)		95	Optimized value
A (time scale) (hrs^b^)		3	Optimized value
B (shape)		0.45 (fast), 1 (fed)	Optimized value
**CR capsule ^3^**			
T (time lag) (h)		0.7	Optimized value
Max (total release) (%)		100	Optimized value
A (time scale) (hrs^b^)		4.7	Optimized value
b (shape)		0.8 (fast), 1.4 (fed)	Optimized value
**Carbamazepine-10,11-epoxide drug-dependent parameters**			
MW (g/mol)		252.27	-
logP		1.4	1.58 [64], 1.97 [64]
Fup (%)		51.8	46.8–51.8 [7]
Blood: plasma concentration ratio (R_bp_)		1.53	1.53 ± 0.45 [53], 1.27–1.80 [54]
pKa		11.03	ADMET Predictor v.10.3.0.0
CL_hepatic_ (L/h)		4.86 ^4^	-
CL_renal,filt_ (L/h)		0.14	0.14 [16]
Peff (cm/s × 10^−4^)		50	50 [43]
Solubility in water (assume pH = 7, mg/mL)		1.34	1.34 [64]
Diffusion Coefficient (cm^2^/s × 10^5^)		0.87	ADMET Predictor v.10.3.0.0
**CBZ-E formulation parameters**			
**Solution and suspension**			
Particle density (g/mL)		1.2	GastroPlus^®^ default value
Mean Particle Radius (µm)		25	GastroPlus^®^ default value
Particle radius standard deviation		0	GastroPlus^®^ default value
Particle radius bin#		1	GastroPlus^®^ default value
**Enteric-coated tablet ^3^**			
T (time lag) (h)		2	Optimized value
Max (total release) (%)		84	Optimized value
A (time scale) (hrs^b^)		1	Optimized value
b (shape)		0.75 (fast)	Optimized value

^1^: Optimized value based on Gerardin 1990 [35] iv infusion subject 1 observed data. ^2^: Optimized value based on Meyer 1992 [36] observed value. ^3^: Using Weibull function to simulate drug dissolution in the intestinal tract. ^4^: Calculated value based on Tomson 1983 [30]; OH-CBZ represents hydroxy carbamazepine; CBZ-glu represents CBZ glucuronide.

### 3.2. CBZ P-M PBPK Model Building and Performance

The CBZ drug-dependent parameters and formulation parameters are listed in Table 1 and Table 2. The local sensitivity analysis indicated that CBZ’s plasma unbound fraction (fu_p_), reference LogD, CL_liver_, and blood-to-plasma ratio (R_bp_) are the key model parameters for the change of the simulated CBZ AUC_0-inf_, with sensitivity values of −1, −0.7, −0.65, and −0.6, respectively (Appendix A). On the other hand, CBZ references LogD, CBZ R_bp_, CBZ-E fu_p_, and CBZ-E CL_liver_ are the key model parameters for the change of the simulated CBZ-E AUC_0-inf_, with sensitivity values of −1.77, 1.63, −0.83, and −0.81, respectively (Appendix A).

The final model accurately recapitulated the observed CBZ concentration–time profiles after administering CBZ as single and multiple doses, and, at the same time, reasonably described the CBZ-E plasma concentration–time curves after CBZ single-dose administration (Appendix A). The model also performs well in capturing the magnitude of the food effect on CBZ single-dose administration (Appendix A). Regarding CBZ PK parameters, 64% (28/44) of simulated AUC_0-inf_ values, 66% (29/44) of simulated AUC_0-t_ values, and 68% (30/44) of simulated C_max_ values were within 80–125% of the observed values. Moreover, 95% (42/44) of simulated AUC_0-inf_ and AUC_0-t_ values, and 98% (43/44) of simulated C_max_ values, were within two-fold of the observed ones (Appendix A). However, the predicted AUC_0-inf_ and AUC_0-t_ values from clinical studies by Kim 2005 [16] and Shahzadi 2011 [65], as well as the C_max_ value from the clinical study by Shahzadi 2011 [65], were outside of the two-fold range. The goodness-of-fit results are illustrated in Figure 3a,b. In total, 45/46 of the predicted plasma concentration AAFE values are within a two-fold range, except for those of Shahzadi’s 2011 clinical study, which had an AAFE value of 3.67 (Appendix A).

The predicted plasma concentration–time profiles of CBZ-E showed discrepancies in comparison to observed ones for some clinical studies following administration of single-dose CBZ, ranging from 200 to 600 mg, under fast and fed conditions and various formulations. In total, 9/13, 7/13, and 12/13 of simulated AUC_0-t_, AUC_0-inf_, and C_max_ values of CBZ-E were within two-fold error of the observed ones. The maximum folds are 2.46, 2.66, and 2.66 for AUC_0-t_, AUC_0-inf_, and C_max_, respectively (Appendix A). Furthermore, 8/13 of AAFE values are within two-fold, as shown in Appendix A. The model tends to overpredict the concentration data, except for the Kim 2005 clinical study [16], which is underpredicted (Figure 3a,b, Appendix A).

The CBZ P-M PBPK model successfully captured the impact of CBZ autoinduction on its plasma PKs when given as multiple doses under fasted conditions. The CBZ-dependent induction parameters are listed in Table 2. In total, 64% (7/11) of simulated AUC_last_ values and 73% (8/11) of simulated C_max_ values for CBZ are predicted within 80–125% of the observed values, while 100% fall within a two-fold error range, using the observed values as the reference (Appendix A). The goodness-of-fit results are presented in Figure 3c,d. Furthermore, 11/11 of AAFE values were within a two-fold range (Appendix A). In summary, the CBZ P-M PBPK model performed well, indicating the successful parameterization of autoinduction across multiple metabolic pathways into the model.

### 3.3. Model Application in DDI

The CBZ-dependent CYP3A4 and CYP2C9 induction parameters are listed in Table 2. The quinidine and dolutegravir PBPK models in GastroPlus^®^ could successfully capture the corresponding drug baselines, whereas the in-house PHT model tended to overpredict PHT exposure, particularly in the initial stages (Figure 4a,b,d, Appendix A). Nevertheless, the PK parameters for PHT are still within a two-fold range of the observed values, a standard in PHT model development and verification [45]. The predicted DDI AUC_last_ and DDI C_max_ ratios for dolutegravir, phenytoin, and tolbutamide are within the prediction success limits proposed by Guest (Figure 4f,g). For quinidine, the DDI AUC_last_ ratio is 0.238, which is within the two-fold range but just outside of the Guest range of 0.249 to 0.70. The predicted DDI for midazolam’s AUC_last_ is 0.103, falling outside of the acceptable two-fold range compared to the observed value of 0.211 (a ratio of 0.49). Meanwhile, the predicted DDI C_max_ ratio for midazolam is 0.183, which falls within the Guest criteria range of 0.18–0.56, derived from the observed ratio of 0.318, as documented in Appendix A. In total, 3/5 DDI AUC_last_ and 5/5 C_max_ ratios meet the Guest criteria (shown in Figure 4f,g).

**Table 2 pharmaceutics-16-00737-t002:** CBZ-dependent induction parameters.

Enzyme	Parameters	Model Value	Literature Value
CYP3A4	E_max_	10(Fit)	4.57–15.73 [66], 15.6 [67], 1.9–14 [68], 6.3–31 [69], 4.7–9.7 [70], 4.1–10.7 [70], 2.3–58 [71], 2.3–40 [72], 9.3–21 [73]; 5.3–11 [74]; 3.48–14.5 [75]; 55.8/60 [76]; 3.75–11.9 [77]
	EC_50, in vitro,T_ (µM)	22(Fix)	14.37–27.70 [66], 58.7/59.1 [67], 4.3–27 [68], 40/42 [69], 13.1–27.2 [70], 10.5–16.2 [70], 12–59 [71], 16 ± 11 [42]; 29–98 [73]; 10.2/34.3 [76]; 21–28 [74]; 47.5 to 80.7 [77]
CYP3A5	E_max_	2.95	2.95 [42], 3.5 [72], 1.43 [78]
	EC_50, in vitro,T_ (µM)	142	142 ± 51 [42]
CYP2C8	E_max_	3.49	3.49 [42]; 3.92 ± 1.34 [79]
	EC_50, in vitro,T_ (µM)	22(Fix)	22 ± 18 [42], 26.62 [79]
CYP2B6	E_max_	10.14	11.5–28 (mean 18.2) [80], 3.7–4.4 [81], 3.08–29.1 [75], 10.14 [42]
	EC_50, in vitro,T_ (µM)	26	9.4–51 (mean 26) [80], 22 ± 9.7 [42]
CYP2C9	E_max_	1.83	1.83 [42]; 2.32 ± 0.33 [79]; 1.0–1.5 [82]
	EC_50, in vitro,T_ (µM)	22(Fix)	30 [42]

E_max_: Maximum induction fold of enzyme activity/expression; EC_50_: the concentration at which 50% of the maximal induction fold is observed.

### 3.4. Nonlinearity PK Investigation

Simulations demonstrate that the AUC_0-inf_/dose decreases as the dose increases when given as a single oral dose, which is consistent with the observed trend (Figure 5a). The red points in the figure represent the observed AUC_0-inf_/dose values calculated from single-dose clinical studies with 200 mg, 400 mg, or 600 mg IR tablets taken under a fasted condition (Appendix A). Two outlier studies, Kim 2005 [16] and Shahzadi 2011 [65], were excluded from the analysis. Single-dose studies with other doses (e.g., 100 mg, 800 mg) were also excluded due to a lack of multiple clinical studies for each.

CBZ exhibits nonlinear dissolution behavior. The commercial IR tablets of CBZ completely disintegrated within 30 s with no agitation [6]. The percentage of drug dissolved in the gut lumen decreases with increasing doses, as shown by the Dis% profiles in Figure 5c, and Fa% follows the same trend. The difference between Dis% and Fa% for the same dose is less than 1% (Appendix A).

It is likely that CBZ has a linear metabolism when given as a single oral dose ranging from 50 to 800 mg, based on the simulation results (Figure 5b). The liver concentrations of CBZ are lower than the Km values of major enzymes CYP3A4 (248 µM or 58.59 µg/mL for CBZ-E production; 282 µM or 66.63 µg/mL for hydroxylated CBZ production) and CYP3A5 (2300 µM or 543.32 µg/mL), as shown in the first 24 h in Appendix A. The enterocyte concentrations of CBZ in the Duodenum and Jejunum1 are close to the Km values for CYP3A4, indicating a potential for saturable metabolism, as shown in the first 24 h in Appendix A. However, the differences between Fa% and FDp% values for all dosages are smaller than 1% and the difference between 50 mg and 800 mg is 0.24%, implying that gut metabolism plays a negligible role in CBZ’s first-pass effect (Figure 5c). This implies that these enzymes are most likely unsaturated in both the enterocytes and the liver, as reflected by the constant CL after single doses for both iv and oral administration (Figure 5b). Additionally, the AUC_0-inf_/dose did not change when given as a single iv dose ranging from 50 to 800 mg, but it decreased when given as a single oral dose, because the Dis% value of CBZ decreases when the dose increases (Figure 5d). In summary, the simulation results suggest that the CBZ’s nonlinear PK is primarily due to the nonlinear dissolution-limited absorption in a single-dose regimen ranging from 50 to 800 mg, resulting in F% decreasing from 95% to 81%.

The nonlinear PKs of CBZ can also be observed when it is administered as multiple oral doses. The simulated Dis% and Fa% curves have only a mild difference compared to those given as a single dose (Figure 5c,d). Similarly, the contribution of gut metabolism to the first-pass effect remains negligible, with a less than 1% difference between Fa% and FDp%. The liver concentrations of CBZ are lower than the Km values of major enzymes CYP3A4 and CYP3A5 (Appendix A). However, the AUC_0-inf_/dose decreases significantly, approximately 2-fold, within the 50–800 mg dose range, while the CL increases 1.64-fold for IR tablets and 1.71-fold for iv infusion, respectively. According to these results, it can be inferred that the nonlinear PK of CBZ observed with multiple doses is attributed to both the decreased absorption due to dissolution limitations and the increased systemic clearance due to autoinduction.

## 4. Discussion

A compressive whole-body P-M PBPK model of CBZ and its active metabolite CBZ-E was successfully established. The model predicted CBZ and CBZ-E plasma concentration–time profiles reasonably well in healthy subjects given 10 to 800 mg single-dose CBZ in different formulations under fasted/fed conditions. The model also successfully captured the autoinduction effect of CBZ in multiple-dose regimens using a maximum effect model. In addition, the model effectively predicted CBZ DDIs with quinidine, dolutegravir, phenytoin, and tolbutamide, all of which fell within the expected two-fold range, although, in the case of midazolam, the DDI AUC_last_ ratio exceeded the two-fold range, indicating a less accurate prediction in this scenario. The model confirmed that CBZ’s absorption in a single oral dose is constrained by its dissolution rate, resulting in nonlinear PK in plasma. In the case of multiple doses, both the nonlinear metabolism caused by autoinduction and the nonlinear absorption due to poor solubility contribute to the observed nonlinear PK behavior.

The CBZ P-M PBPK model performed well in predicting CBZ plasma concentrations, except for two clinical studies, Kim 2005 [16] and Shahzadi 2011 [65]. The predicted PKs for these studies fell outside of the two-fold range shown in Figure 3a,b and Appendix A. However, when all clinical observed data were plotted together, it was clear that these two studies were not compatible with the other eight clinical studies that used the same dose regimen (200 mg) for the same drug formulation (IR tablet), indicating that these two studies were outliers. The summary plot can be found in Appendix A.

We validated the major pathways by comparing their contributions to literature values. In our model, CYP3A4, CYP3A5, and CYP2C8 contribute 37.68% to CBZ-E formation, consistent with literature values of 20–40% (inhibition of CBZ-E formation by triacetyloleandomycin in human liver microsomal incubation) and 41–45% (by CYP3A4 IgG) [13]. The production of OH-CBZ is a minor metabolic pathway primarily catalyzed by CYP2B6 and CYP3A4 for 3-OH-CBZ formation, with multiple CYPs involved in 2-OH-CBZ formation [11]. To simplify the PBPK model, we included only CYP2B6 and CYP3A4. Our model indicates that 42.98% of CBZ is converted into CBZ-E and hydrolyzed CBZ, aligning with literature values of 39.2% ± 9.8% [86]. This supports the model’s accuracy in capturing the major pathway for CBZ to CBZ-E conversion and the overall CYP3A pathway, with potential for future updates as more enzyme data become available.

The model tends to overpredict the CBZ-E plasma concentration, except for that of the clinical study by Kim 2005 [16], which is underpredicted. Several factors likely contribute to this discrepancy. Firstly, the absence of information on enzyme polymorphisms (CYP3A4, CYP3A5, and EPHX1) could be significant, as CYP3A4 and CYP3A5 play significant roles in CBZ-E production, and CBZ-E is almost exclusively metabolized via EPHX1. Zhu et al. [87] reported a lower concentration: dose ratio (CDR) of CBZ-E in CYP3A4*1G variant carriers (with enhanced enzyme activity) compared to the CYP3A4*1/*1 carriers, consistent with observations by Zhao et al. [88] via meta-analysis, but Ma et al. [89] found no significant differences. Miyata-Nozaka et al. [90] reported that CYP3A5*3 carriers (associated with decreased CYP3A5 enzyme activity) exhibit a negligible effect on CBZ-E, but they highlight the need for a larger donor pool in order to demonstrate the effects of CYP3A5 polymorphism, as CYP3A5*3 carriers have a high CDR_CBZ_ [87,91,92]. EPHX1 c.416A>G variants (associated with decreased enzyme activity [93]) and c.128G>C variants (associated with increased enzyme activity [94]) showed substantial correlations with the CBZ-diol: CBZ-E ratio [87,95]. Patients with the EPHX1 c.337T>C CC genotype exhibited higher CBZ-E concentrations than TT genotype carriers [89,96]. Consequently, the lack of enzyme activity details in the clinical studies used for model development, stemming from missing volunteer information, could contribute to the disparity between CBZ-E observations and predictions. In addition, CBZ-E is a substrate for the P-gp transporter [97], with ABCB1 3435CT carriers showing significantly lower CDR_CBZ-E_ compared to ABCB1 3435CC carriers [87]. However, our model does not incorporate the transporter parameters due to their unavailability.

We assumed that EC50_CYP3A4_ = EC50_CYP2C8_ = EC50_CYP2C9_ = 22 µM to simplify the description of carbamazepine (CBZ)’s induction effect on CYP3A4, CYP2C8, and CYP2C9 enzymes. The reported values of EC_50_,_CYP3A4_ and E_max,CYP3A4_ are highly variable, with an EC_50_ range of 4.3 to 98 µM, an E_max_ value ranging from 1.9 to 60 (Table 2), and E_max_/EC_50_ ratios ranging from 0.07 to 1.03 (Appendix A). To determine these values, we reviewed the induction mechanisms of CBZ on the enzymes in question. CBZ primarily induces CYP3A4 via the pregnane X receptor (PXR) [98,99], as confirmed by the FDA DDI draft guidance [28]. However, there is some discrepancy in the literature regarding the induction of CYP2C enzymes. While the FDA DDI draft guidance states that both CYP3A4 and CYP2C enzymes are induced via PXR activation [28], Zhang et al. [42] reported that the induction profiles of CYP2C and 3A enzymes can substantially differ depending on the compound. Nagai et al. [79] also concluded that CYP2C induction is regulated by both PXR and CAR, with varying contributions among CYP2C forms. Nonetheless, we assumed that EC50_CYP3A4_ = EC50_CYP2C8_ = EC50_CYP2C9_ = 22 µM, based on the following evidence: (1) the average value of EC50_CYP3A4_ is 22.13 µM, based on enzyme activity measurement (Appendix A); (2) the fold-induction values of CYP2C8 and CYP2C9 were well-correlated with those of CYP3A4 (r = 0.895 and 0.886 for CYP2C8 and CYP2C9, respectively) [79]; (3) there was a positive correlations for induction between CYP3A4 and CYP2C mRNA (with r^2^ ranging from 0.6 to 0.99 [100]); and (4) the differences among the reported EC_50_ values were not significant (16, 22, and 30 μM for CYP3A4, CYP2C8, and CYP2C9, respectively) [42]. After fixing EC50_CYP3A4_, we optimized Emax _CYP3A4_ to fit the clinical data. The final Emax value for CYP3A4 is within the range of in vitro experimental data based on the enzyme activity analysis [66,69,71].

The model’s prediction of the DDI AUC_last_ ratio for midazolam (0.49) falling outside of the standard two-fold range suggests a potential overestimation of CBZ’s induction effect on CYP3A4. Several factors could contribute to this discrepancy. The midazolam PBPK model used in the study only considers the CYP3A4 enzyme and does not account for CYP3A5 gene polymorphism. However, it is important to consider the contribution of CYP3A5, particularly in CYP3A5*1 (wildtype allele) carriers. The frequency of the CYP3A5*1 wildtype allele in the white population ranges from 0.12 [101] to 0.15 [102], and 65.9% of the study participants were white. In CYP3A5*1 carriers, CYP3A5 constitutes greater than 50% of the total CYP3A content in the liver and jejunal epithelium [101]. In addition, purified CYP3A5 displayed a two-fold greater rate of 1-hydroxymidazolam formation and a similar rate of 4-hydroxy midazolam formation in a reconstituted system compared to CYP3A4. [103]. Consequently, in CYP3A5*1 carriers, CYP3A5 contributes more than 50% to midazolam metabolism when given midazolam without CBZ. Moreover, CBZ demonstrates a stronger induction capacity for CYP3A4 than CYP3A5 (shown in Table 2), with an enzyme induction fold of 4 for CYP3A4 compared to 1.2 for CYP3A5 after administration of 300 mg of CBZ twice a day for 18 days (Appendix A). This suggests that a significant difference in midazolam PK might exist between CYP3A5*1 carriers and non-carriers when co-administered with multiple doses CBZ. However, the lack of CYP3A5 genetic information for the study volunteers limits our ability to fully understand the impact of midazolam metabolism and its potential interaction with CBZ in individuals carrying the CYP3A5*1 allele. Nonetheless, the predicted CYP3A4 enzyme induction fold remains consistent with reported values ranging from 2.74 to 4.05 folds. These values are based on the detection of CYP3A4 gene expression changes in the livers of epileptic patients who received a daily dose of 600 mg of CBZ compared to those who did not (with all livers being homozygous non-expressers of CYP3A5) [104]. Therefore, the CYP3A4 induction parameters were not further refined, as the model accurately predicts the DDI AUC_last_ and C_max_ ratios for quinidine and dolutegravir as well as the DDI C_max_ ratio for midazolam. Moreover, the DDI AUC_last_ ratio of 0.49 is very close to the commonly used criteria threshold of 0.5, and the model correctly captures the autoinduction effect. Further validation with additional sensitive CYP3A4 substrates will be needed in the future.

The predicted DDI ratios of AUC_last_ and C_max_ for the PHT and tolbutamide are within the Guest criteria (see Equations (5)–(7)) of the observed values, demonstrating a well-described induction effect of CBZ on enzyme CYP2C9. After 18 days of CBZ administration at 300 mg twice daily, the enzyme induction fold for CYP2C9 was 1.2, which is consistent with the range of 1.35 to 1.54 observed in enzyme protein induction folds (digitized data) [105]. However, it is important to note that the model did not predict the CBZ impact on PHT PK when single doses of the two drugs are administered simultaneously. The PHT clinical study consists of three phases [85], where subjects received either a single dose of 600 mg of PHT (Phase I), a single dose of 600 mg of PHT along with a simultaneous single dose of 400 mg of CBZ (Phase II), or a single dose of 600 mg of PHT and a dose of 400 mg of CBZ taken at the same time after taking 200 mg of CBZ twice a day for 7 days (Phase III). The PHT PBPK model accurately captured the PHT concentration–time profiles of Phase I, with predicted versus observed ratios of 1.33, 1.17, 1.23, and 0.5 for AUC_0-inf_, AUC_0–72_, C_max_, and t_max_, respectively (see Appendix A), indicating its potential for accurately predicting PHT PK characteristics. However, when PHT was administered with CBZ at the same time in Phases II and III, the observed t_max_ reduced from 6 h to 4 h, and the observed C_max_ reduced from 5.85 (for Phase I) to 4.79 µg/mL (for Phase II) and 4.57 µg/mL (for Phase III), as shown in Appendix A. The mechanism behind this change is currently unknown. To address this limitation, the PHT concentration–time profile in Phase II was used as a baseline to exclude any factors other than CYP2C9 induction, under the assumption that the CYP2C9 induction extent was negligible for a single dose of CBZ. Therefore, the difference in PHT PKs between Phase II and III could be attributed to the induction effect of multiple-dose CBZ administration on CYP2C9.

CBZ exhibits dissolution-rate limited absorption in a single oral dose due to its poor solubility, which leads to nonlinear PK. The CBZ’s nonlinear PK exploration plot (Figure 5a) in the single oral dose regimen showed that the observed values had a steeper slope than the predicted ones, indicating either a higher clearance or a lower fraction absorbed at a dose of 600 mg. However, assuming negligible enzyme induction, a higher clearance may not be a reasonable explanation, as all of the clinical studies used a single dose of CBZ. The ratios of AUC_0-inf_/dose exhibited a large variability (0.57–1.11 for 200 mg in nine studies, 0.60–0.91 for 400 mg in seven studies, and 0.36–0.64 for 600 mg in four studies). Therefore, this discrepancy at 600 mg could be due to large interstudy variability and an insufficient number of available studies at a 600 mg dose. Comparisons to two single-dose escalating studies revealed that the results of Gerardin 1976 (100–600 mg, tablet) [19] were in line with the predicted trend, while Cotter 1977 (200–900 mg, tablet) [18] showed a contradictory trend above 500 mg, suggesting an increasing in F or a decrease in Ke (shown in Appendix A). Given the large variability, a single clinical study may not accurately reflect the real trend. In combination with a parallel elimination rate observed by Levy 1975 [6] for 3 to 9 mg/kg of CBZ propylene glycol solution, it is most likely that CBZ has a linear clearance for a single dose up to 600 mg. Therefore, CBZ’s nonlinearity for a single dose is more likely caused by nonlinear absorption in the dose range of 50 to 600 mg. However, because there are not multiple clinical studies at higher single doses, it is unclear whether CBZ saturation of enzymes occurs at doses greater than 600 mg.

There is an ongoing debate regarding whether CBZ is a substrate of P-glycoprotein (P-gp) and multi-drug resistance protein 2 (MRP2). Some studies suggest that P-gp is not involved in CBZ disposition, including those conducted by Zhang et al. [97] (using polarized cell lines) and Owen et al. (working with mdr1a/1b(-/-) and wildtype mice, Caco-2 cells, and flow cytometry in lymphocytes using rhodamine 123) [106]. However, Potschka et al. [107] demonstrated that CBZ is a substrate for both P-gp and MRP2 using in vivo micro dialysis local perfusion with verapamil in animal models. Nevertheless, Ferreira et al. [108] suggested that the changes observed in CBZ PK after pretreatment with verapamil are caused by CYP3A4 inhibition rather than P-gp involvement. Additionally, Radish et al. [109] found that CBZ is not a substrate for human MRP2 using ABCC2-transfected cell lines. Furthermore, no clinically significant association was observed between MRP2 genetic variants and CBZ treatment outcomes, suggesting that CBZ may not be a substrate of P-gp or MRP2. Instead, Awasthi et al. [110] reported that CBZ is a substrate for RLIP76, a non-ABC multi-specific transporter expressed in brain tissue, preferentially in the luminal surface of endothelial cell membranes, and with a significant overlap in substrates with P-gp. It is also worth noting that CBZ induces the expression of P-gp mRNA and MRP2 mRNA, but it is unclear whether it can elevate their protein abundance. Giessmann et al. [111] observed that after repeated administration of 600 mg of CBZ once daily for 18 days, the intestinal expressions of the P-gp mRNA, MRP2 mRNA, and protein content of MRP2, but not of P-gp, were significantly increased. Brueck et al. [112] found that CBZ induced the mRNA expressions of P-gp and MRP2, but it did not increase protein abundance. In contrast, Jarvinen et al. [105] reported an upregulated protein level of P-gp, MRP2, and human organic cation uptake transporter (OCT1) using a 3D spheroid primary human hepatocytes approach. Our model did not incorporate any transporter information due to conflicting evidence and the unavailability of in vitro parameters.

Several published studies have utilized PBPK modeling to investigate the PK of CBZ in pediatric [62] and pregnant [113] populations, describe gastrointestinal absorption of various CBZ formulations [37], and explore DDIs [23,43,67,114,115,116]. The existing models, such as those established in PK-Sim^®^ [117] and Simcyp^TM^ [67], simplified the absorption process of CBZ by using the Weibull function and a first-order absorption model, respectively. Consequently, these models lack a comprehensive mechanism-based approach to understand absorption. On the other hand, the models previously developed in GatroPlus^®^ have utilized the ACAT™ model to incorporate a full mechanism-based approach. However, they have simplified the systemic drug disposition process either by using a compartmental PK model for representation [37] or by considering only the CYP3A4 enzyme [62]. Moreover, the previously published models developed in GastroPlus^®^ have not accounted for the autoinduction effect of CBZ, rendering them inadequate for multiple dose regimens. To address these limitations, we have leveraged the granular nature of the ACAT^TM^ model in conjunction with a whole-body PBPK model. This combined approach allows for a detailed analysis of the factors that contribute to the nonlinear PKs of CBZ, encompassing both absorption and metabolism. The CYP3A5 enzyme was added into CBZ metabolism pathways in our model, in addition to CYP3A4, CYP2C8, CYP2B6, and UGT2B7. Both CYP3A4 and CYP3A5 are expressed in the liver and intestine, with CYP3A5 being the predominant form expressed in extrahepatic tissues. Notably, we also evaluated the effect of CBZ’s induction effect on the enzyme CYP2C9, a previously unexplored aspect of its pharmacokinetics.

Our model accurately described the CBZ autoinduction effect, but we encountered the inability of the DDI module to incorporate the CBZ autoinduction effect when CBZ acts as a victim drug. Additionally, we faced challenges including CBZ-E in the multiple-dose simulation due to a lack of EPHX1 enzyme activity data, which are necessary for understanding both CBZ-E metabolism (V_max_ and K_m_) and CBZ’s induction effect (E_max_ and EC_50_) on EPHX1 [86]. CBZ-E’s concentration-dependent activation of PXR suggests it also induces CYP3A, adding another layer of complexity to CBZ’s metabolic profile [104]. The available evidence suggests that both CBZ and CBZ-E likely have an effect on the PXR pathway [104], and the fitted Emax value for CYP3A4 induction by CBZ may also be compensating for the missing induction effect of CBZ-E. We anticipate overcoming these limitations as more experimental data relatehd to the impact of both CBZ and CBZ-E on the enzymes become available in the future.

## 5. Conclusions

A whole-body parent–metabolite PBPK model of carbamazepine was successfully established in GastroPlus^®^. The model demonstrated accurate capture of the autoinduction effect of carbamazepine and its ability to predict plasma concentration–time profiles for both single- and multiple-dose regimens with various formulations. Additionally, the model can help to understand potential Drug–Drug Interactions when carbamazepine is used as a CYP3A4 and CYP2C9 inducer. By incorporating the Advanced Compartmental Absorption and Transit (ACAT^TM^) model, we were able to identify the nonlinear pharmacokinetic behavior of carbamazepine for multiple-dose regimens, which is due to its nonlinear absorption and autoinduction properties. This PBPK model holds significant promise in supporting drug development by enabling designs of clinical trials that minimize the risk of side effects and potentially informing drug-label information in untested clinical scenarios.

## Figures and Tables

**Figure 1 pharmaceutics-16-00737-f001:**
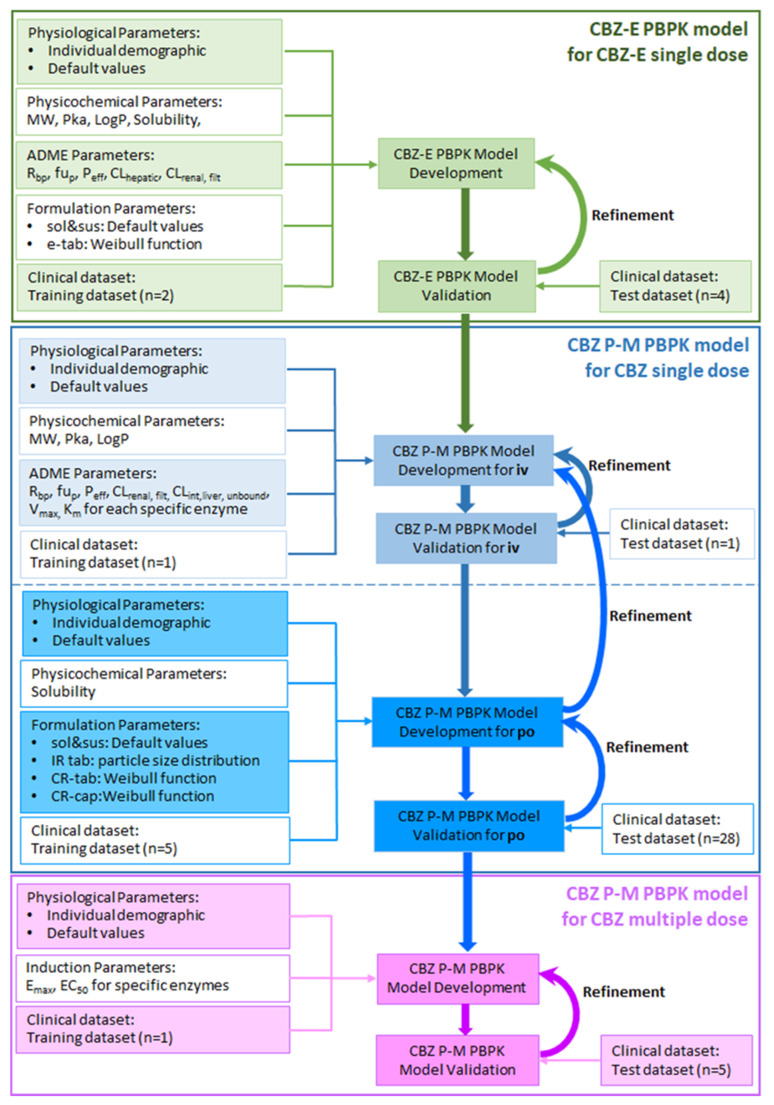
The workflow of the CBZ P-M PBPK model’s development and verification. MW: molecular weight; pKa: acid dissociation constant; LogP: octanol-water partition coefficient; R_bp_: blood to plasma ratio; fu_p_: fraction unbound in plasma; P_eff_: effective permeability; CL_hepatic_: hepatic clearance; CL_renal,filt_: renal clearance for unbound drug; CL_int,liver,unbound_: unbound hepatic intrinsic clearance; V_max_: enzyme maximum metabolism rate; Km: Michaelis–Menten constant; E_max_: maximum induction fold of enzyme activity/expression; EC_50_: the concentration at which 50% of the maximal induction fold is observed; iv: iv infusion; sol: solution; sus: suspension; e-tab: enteric-coated tablet; IR tab: immediate-release tablet; CR-tab: control/extended/sustained-release tablet; CR-cap: control/extended/sustained-release capsule; iv: intravenous administration; po: oral administration.

**Figure 2 pharmaceutics-16-00737-f002:**
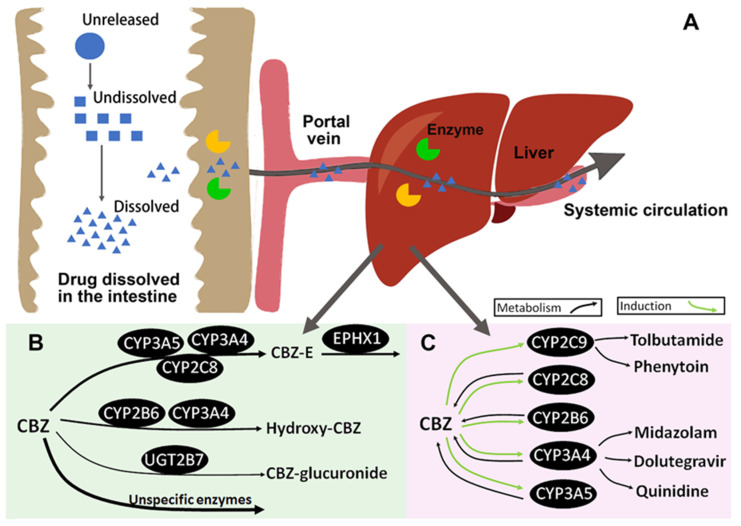
The schematic illusion of the CBZ P-M PBPK model structure. (**A**) The summary plot of the CBZ drug dissolution process and first-pass effect; (**B**) CBZ metabolic pathways. The width of the black arrows shows the corresponding contribution to the total drug elimination; (**C**) CBZ autoinduction and DDI network of CYP3A4 and CYP2C9.

**Figure 3 pharmaceutics-16-00737-f003:**
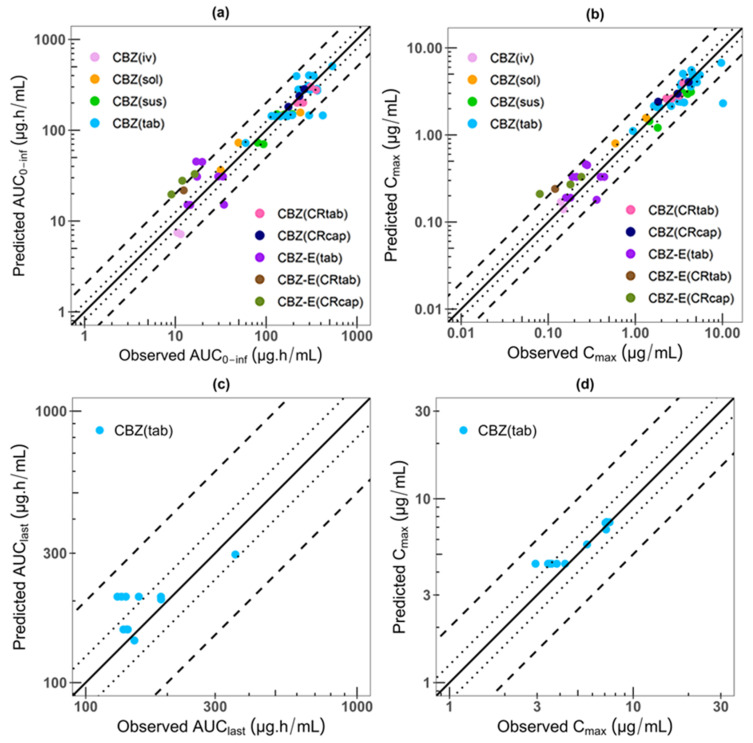
Performance of the CBZ P-M PBPK model. Predicted compared to observed (**a**) AUC_0-inf_ and (**b**) C_max_ values after oral administration of CBZ single dose. Predicted compared to observed (**c**) AUC_last_ and (**d**) C_max_ values after oral administration of CBZ multiple doses. The line of identity is shown as a solid line, 1.25-fold deviation is shown as a dotted line, and 2-fold deviation is shown as a dashed line.

**Figure 4 pharmaceutics-16-00737-f004:**
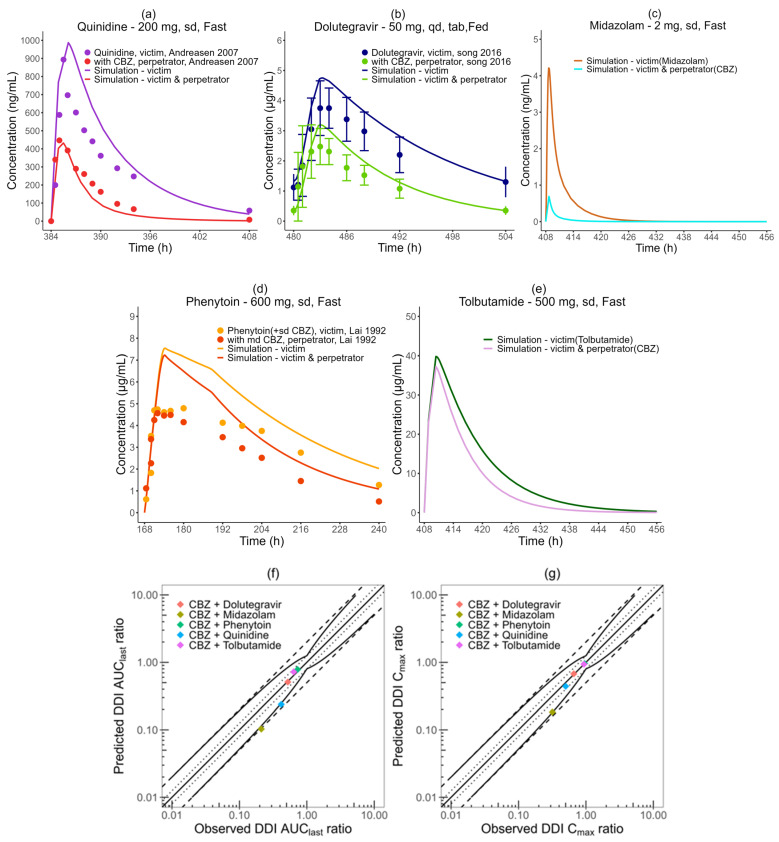
DDI prediction performance of the CBZ P-M PBPK model. Upper and center rows: predicted plasma concentration–time profiles of (**a**) quinidine, (**b**) dolutegravir, (**c**) midazolam, (**d**) phenytoin, and (**e**) tolbutamide with and without co-administration of carbamazepine in comparison to observed data (plots a [83], b [84], d [85]). Bottom row: goodness-of-fit plots of predicted versus observed (**f**) DDI AUC_last_ ratios and (**g**) DDI C_max_ ratios. A straight, solid line represents the line of identity, while dotted lines indicate a 1.25-fold deviation. A 2-fold deviation appears as dashed lines. Curved, solid lines delineate the prediction success limits proposed by Guest et al. [46].

**Figure 5 pharmaceutics-16-00737-f005:**
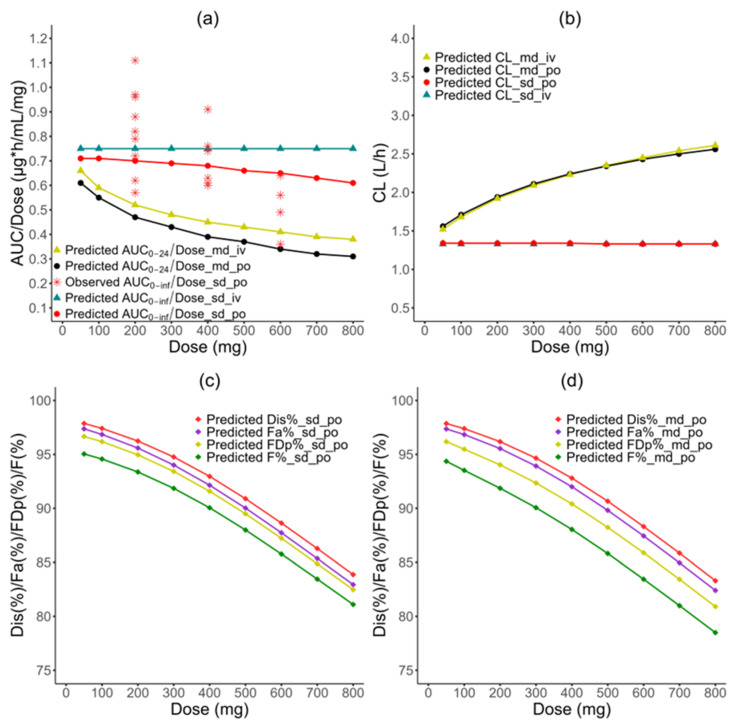
Carbamazepine nonlinear PK exploration results. The relationship between normalized AUC (AUC/dose) and doses (**a**), CL and doses (**b**), Dis% and dose, Fa% and doses, FDp% and doses, as well as F% and doses in CBZ single- (**c**) and multiple-dose regimens (**d**). CL: total systemic clearance; Dis%, Fa%, FDp%, and F are represented as the percentage of drug dissolved, in enterocytes, in the portal vein, and in systemic circulation, respectively.

## Data Availability

Data are contained within the article and Appendix A.

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
