# Peer review of "Applying Physiologically Based Pharmacokinetic Modeling to Interpret Carbamazepine’s Nonlinear Pharmacokinetics and Its Induction Potential on Cytochrome P450 3A4 and Cytochrome P450 2C9 Enzymes"

_pharmaceutics, 2024, doi:10.3390/pharmaceutics16060737_

Round 1
Reviewer 1 Report
Comments and Suggestions for Authors
This is a very informative paper which is currently very much of interest due to the consideration of carbamazepine as a possible index inducer. The model is well developed and validated and is fully mechanistic for absorption and drug disposition, this is appreciated.
The reviewer would strongly recommend its publication with just a few minor comments in terms of detail in the method section and the ultimate claim in the abstract to be amended (see below).
In the method, it is not clear that gut metabolism has been included e.g. in the text associated with figure 1, only hepatic metabolism is mentioned (line 158) however the discussion suggests that this is the case. It is also not clear why the different dose forms are mentioned as they do not appear to feature in the figure.
It appears that 56% of the metabolism is not specified (line 217), this is a concern.
A parameter range of =/- 20% for local sensitivity seems low (line 278).
It is considered that the model allows an important understanding of the PK of carbamazepine and its induction. However there is some mis-prediction e.g. in single dose carbamazepine profiles (some more than 2 fold) (line 363) and the reviewer considers that 2 fold is lenient. In particular it is a concern that midazolam (often used as the index substrate for CYP 3A4) does not predict within 2 fold and that there are no clinical profiles to compare. Considering this, it is suggested that the claim in the abstract that this model can now be used 'to predict induction' is not supported. It is proposed that this be reworded to something more cautious e.g. In further applications the model can be used to support the understanding of DDIs. It would not be agreed that is could be used to accurately predict the extent of induction for a novel drug with a high contribution of CYP3A4/2C9 metabolism.
The wording in the discussion is considered appropriate and the content is excellent.
Reviewer 2 Report
Comments and Suggestions for Authors
The manuscript describes development and validation of PBPK for carbamazepine and its metabolite, for the different dosage forms of this drug. The authors made an organized and extensive analysis of the previously published clinical pharmacokinetic data of carbamazepine, and other data that are relevant to this drug, including the dosage forms, absorption, transport and metabolic pathways, etc. The developed model was extensively validated based on the concentration vs. time datasets of carbamazepine and metabolite, and on clinical DDI data for carbamazepine with other drugs. The manuscript is well organized and well written. The tables and figures are informative and of high quality. I recommend to publish this manuscript, and suggest the authors to use the developed model for exploration/prediction of the clinical PK of carbamazepine and its DDIs in the individual patients (with different CYP3A4 & 2C9 genotype and phenotype), with specific disease states (e.g., different extent of the liver disease, Child Pugh A-C), with different patterns of drug adherence.
Minor comments:
1. Please consider to add some quantitative data from Fig. 5 to the Abstract and Conclusions sections of the manuscript.
E.g., in the range of the therapeutic doses (50-800 mg), the oral bioavailability decreases with the increase in the dose from 95% to 80%, in non-linear fashion.
In multiple dosing, the clearance increases in non-linear fashion with the increase in the dose. The overall extent of clearance induction within the 50-800 mg dose range is approximately 2-fold.
2. The simulated/predicted PK data in the figures (e.g., Fig 4a-e, Fig. S4) are not smooth, and appear to be based on a small number of time points. Please check the time step of these simulations, and consider to shorten it.
3. Line 181- “The development … was developed …” – please correct the grammar of this sentence.
